# Load forecasting method based on CEEMDAN and TCN-LSTM

**Luo Heng** [1,2]*, **Cheng Hao**[1], **Liu Chen Nan**[1]

**1** School of Electronics and Information Engineering, University of Science and Technology of Suzhou, Suzhou, Jiangsu, China, **2** Key Laboratory of Intelligent Energy Saving in Buildings, University of Science and Technology of Suzhou, Suzhou, Jiangsu, China

* 1727938264@qq.com

**Data Availability Statement:** All data are in the manuscript and/or Supporting information files.

**Funding:** The author(s) received no specific funding for this work.

## Abstract

Aiming at the problems of high stochasticity and volatility of power loads as well as the difficulty of accurate load forecasting, this paper proposes a power load forecasting method based on CEEMDAN (Completely Integrated Empirical Modal Decomposition) and TCN-LSTM (Temporal Convolutional Networks and Long-Short-Term Memory Networks). The method combines the decomposition of raw load data by CEEMDAN and the spatio-temporal modeling capability of TCN-LSTM model, aiming to improve the accuracy and stability of forecasting. First, the raw load data are decomposed into multiple linearly stable subsequences by CEEMDAN, and then the sample entropy is introduced to reorganize each subsequence. Then the reorganized sequences are used as inputs to the TCN-LSTM model to extract sequence features and perform training and prediction. The modeling prediction is carried out by selecting the electricity compliance data of New South Wales, Australia, and compared with the traditional prediction methods. The experimental results show that the algorithm proposed in this paper has higher accuracy and better prediction effect on load forecasting, which can provide a partial reference for electricity load forecasting methods.

## 1 Introduction

With the continuous expansion of the power system scale and the increasing power load, accurate power load forecasting has become more and more important [1]. Power load forecasting is important for the dispatch and operation of power systems, as well as the basis for power market regulation, power supply and demand balance, and energy planning [2]. Accurate forecasting of power load trends and changes is crucial for electric utilities and energy suppliers because they need to develop reasonable power dispatch and resource allocation strategies based on the forecast results to ensure the stability and reliability of the power system [3]. In addition, accurate power load forecasting is also a key factor in facilitating renewable energy integration, energy trading markets and energy efficiency [4–6], which can improve economic and social benefits.

Traditional power load forecasting methods mainly include linear regression analysis, time series analysis, and BP neural network. However, these methods have their own shortcomings,

**Competing interests:** The authors have declared that no competing interests exist.

such as insufficient adaptation to nonlinear relationships and low prediction accuracy and robustness [7, 8]. In recent years, with the rapid development and application of deep learning technology, power load forecasting methods based on deep learning have also received widespread attention. Among them, models such as Convolutional Neural Networks (CNN) and Long Short-Term Memory Networks (LSTM) have distinctive advantages and are widely used in time series forecasting. Feng [9] et al. used a convolutional neural network based approach to predict power loads and proved in experiments that it has better prediction performance than traditional methods. Mao [10] et al. proposed a new approach based on long and short term memory networks to capture long term dependencies in power load data. Jiang et al. [11, 12] attempts to apply deep learning models to solve some challenging problems in power load forecasting.

In addition, some scholars have also proposed prediction models that combine sequence decomposition algorithms and machine learning algorithms. In the algorithm for time series decomposition, commonly used are wavelet transforms [13], Empirical Modal Decomposition [14] and Collective Empirical Modal Decomposition [15]. The purpose of the models combining these decomposition algorithms and machine learning algorithms is to transform a nonlinear, nonsmooth time series into multiple, relatively smooth subsequences, and to improve the prediction of the entire model by predicting the multiple subsequences individually. While decomposition of a time series is not a necessary step for time series forecasting, but in forecasting studies in areas such as finance and wind speed [16, 17], it has been widely used and achieved better prediction results. Zhang et al. [18] utilizes empirical modal decomposition to decompose the original load sequence into multiple intrinsic modal functions (IMFs), and then the IMFs are used as inputs to the LSTM model for load prediction, which effectively improves the prediction accuracy, and can show better performance especially in the case of nonlinear and nonsmooth load data. An improved short-term load forecasting model based on EMD and LSTM was proposed by Zhao et al. [19]. The experimental results show that the model has significant effect in improving the prediction accuracy and robustness. Qin [20] proposed a combined model based on EMD and Extreme Learning Machine (ELM). The experimental results show that the proposed combined model can better adapt to nonlinear and nonsmooth load data and has high prediction accuracy. Luo [21] introduced a regional electric load forecasting method based on empirical modal decomposition and support vector regression (SVR). First, the original load sequence is decomposed into multiple IMF components using EMD, and then these components are utilized to forecast future loads, which are combined by the SVR model. The experimental results show that the proposed method can better adapt to nonlinear and nonsmooth load characteristics and improve the prediction accuracy. Huang [22] et al. proposed a power load forecasting method using empirical modal decomposition, least squares support vector regression (LS-SVR), and firefly algorithm, which greatly improves the accuracy and stability of the forecast.

In order to improve the prediction accuracy on nonlinear and unsteady power loads, this paper proposes a combined prediction model with adaptive complete modal empirical decomposition, temporal convolutional network, and long-short-term memory network (CEEMDAN-TCN-LSTM). The method combines empirical modal decomposition and deep learning models, aiming to overcome the limitations of traditional deep learning methods and improve the accuracy and stability of power load forecasting. First, we use an empirical modal decomposition algorithm to decompose the raw power load data into multiple intrinsic modal functions (IMFs). Then, we take the decomposed IMFs as inputs and construct a TCN network to extract the spatio-temporal features of each IMF. In this paper, we introduce the attention mechanism in the TCN network, so that the TCN can better solve the interdependence

between complex variables as a way to obtain obtain a more accurate feature representation. Next, we employ a long and short-term memory network to model the temporal dependencies of IMFs. Finally, we realized the integration of CEEMDAN and TCN-LSTM models by stacking and jointly training TCN and LSTM networks. Through the combination of multilevel and nonlinear structures, the prediction results obtained from each sequence are stacked to obtain the complete prediction value of the original sequence. In this paper, the combined CEEMDAN-TCN-LSTM model is compared and experimented with traditional deep learning methods and classical time series models. The experimental results show that the method has better accuracy and stability in power load forecasting compared to traditional deep learning methods and time series models.

## 2 Materials and methods

### 2.1 Empirical modal decomposition of complete ensembles of adaptive noise

CEEMDAN is an improved decomposition method based on EEMD, which is essentially an EMD decomposition of a signal several times and suppresses the generation of modal aliasing by superimposing Gaussian white noise to change the polar point characteristics of the signal. However, EEMD cannot completely remove the noise during the decomposition process, and there exists noise residue. To solve this problem, CEEMDAN introduces the concept of adaptive noise. In CEEMDAN, the variance of the noise is obtained by adaptive estimation of each IMF instead of a fixed Gaussian white noise [23]. By incorporating adaptive white noise, CEEMDAN is able to completely separate the eigenmode functions and also reduces the reconstruction error. The CEEMDAN algorithm steps are as follows:

(1) Add Gaussian white noise to the original signal sequence to obtain a new sequence as:

$$x^i(t) = x(t) + \varepsilon\omega^i(t), i = 1, 2, \ldots, n \tag{1}$$

Formula: $x(t)$ is the original signal; $x^j(t)$ is a new signal; $\varepsilon$ is the signal-to-noise ratio between signal and noise; $i$ is the number of times to add Gaussian white noise; $\omega^i(t)$ is the added Gaussian white noise.

(2) Perform EMD decomposition on the resulting new signal sequence and average the first value of the decomposition as the first IMF component obtained by CEEMDAN decomposition.

$$C_1(t) = \sum_{i=1}^{n} E_1^i(t) \tag{2}$$

Formula: $C_1(t)$ is the first IMF component produced by the CEEMDAN decomposition; $E_1^i(t)$ is the $i$th IMF component obtained after EMD decomposition. The first residual component is then obtained:

$$r_1(t) = x(t) - C_1(t) \tag{3}$$

(3) Add Gaussian white noise to the first residual component signal obtained after decomposition and perform EMD decomposition.

$$C_j(t) = \frac{1}{n} \sum_{i=1}^{n} E_1 \left( r_{j-1}(t) + \varepsilon E_{j-1}(x_i(t)) \right) \tag{4}$$

$$r_j(t) = r_{j-1}(t) - I_j(t) \tag{5}$$

Formula: $C_j(t)$ is the $j$th IMF component obtained from the CEEMDAN decomposition; $E_{j-1}$ is the $j-1$ th IMF component of the EMD decomposition; $r_j(t)$ is the residual component after the $j$th decomposition.

(4) Repeat the above steps until the residual component can no longer be decomposed, i.e., the CEEMDAN decomposition is finished. The original signal sequence is decomposed into several IMF components and a residual component as follows:

$$x(t) = \sum_{i=1}^{n} C_i(t) + r(t) \tag{6}$$

Formula: $C_i(t)$ is the $i$th IMF component; $r(t)$ is the residual component.

## 2.2 Sample entropy

Sample entropy is an algorithm that measures the degree of disorder and randomness of a sequence, and this algorithm is widely used for quantitative detection of sequences containing noise [24]. The smaller value of sample entropy indicates the higher degree of disorder of the sequence, and vice versa indicates the better regularity of the time series, so it is suitable for the complexity analysis of the load data after CEEMDAN decomposition. Sample entropy, as an improved algorithm of approximate entropy, has better consistency effect and faster calculation speed, and its calculation process is as follows:

(1) Reconstruct the original signal sequence of length N into a vector of dimension:

$$X(i) = \{x(i), x(i+1), \cdots, x(i+m-1)\}, (i = 1, 2, \cdots, N-m+1) \tag{7}$$

(2) Calculate the distance between the lost vector and the other vectors:

$$d_{ij} = \max|x(i+k) - x(j+k)|, (k = 0, 1, 2, \cdots, m-1) \tag{8}$$

(3) Setting the threshold value $r$, get the number $n_{ij}(r)$ of $d_{ij}$, calculate its ratio to the number of vectors $N - m+1$, denoted as $C_i^m(r)$.

$$C_i^m(r) = \frac{n_{ij}(r)}{N - m + 1} \tag{9}$$

It is worthwhile to take an average for it:

$$C^m(r) = \frac{1}{N - m + 1} \sum_{i=1}^{N-m+1} C_i^m(r) \tag{10}$$

(4) Increase the dimension to $m+1$ and repeat the above steps to get the mean:

$$C^{m+1}(r) = \frac{1}{N - m} \sum_{i=1}^{N-m} C_i^{m+1}(r) \tag{11}$$

(5) When the length N of the time series is a finite value, define the sample entropy as:

$$SampEn(m,r) = -\ln \frac{C^{m+1}(r)}{C^m(r)} \tag{12}$$

## 2.3 Temporal convolutional network

Temporal Convolutional Network (TCN), which is improved based on Convolutional Neural Network (CNN), is a kind of neural network that can be used to deal with the structure of time series [25]. Compared to the traditional CNN, the feedback and convergence speed of the TCN model is further improved, and the learning ability of the network is further enhanced.

Causal dilation convolution is the core part of TCN network. In causal dilation convolution, the parameters such as convolution kernel, number of convolution layers and dilation coefficients can be adjusted to realize the feature extraction of the time series as a whole and obtain the temporal dependence. The network structure of TCN is shown in Fig 1.

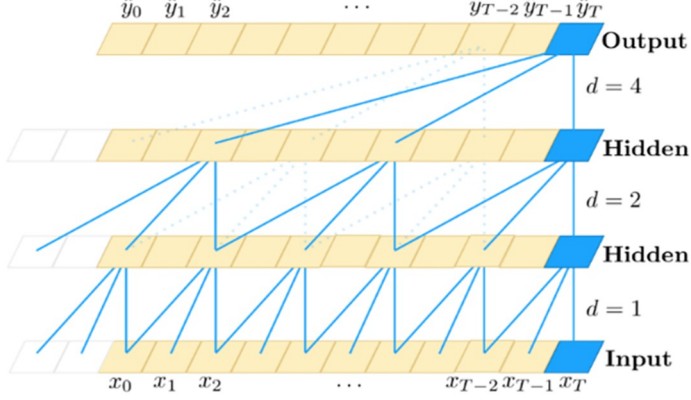

**Fig 1. TCN network structure diagram.**

The causal dilation convolution is sampled with the dilation rate $d$ as the base interval, and the bottom layer $d = 1$ denotes that the feature extraction is performed on the value of each input network; $d = 2$ denotes that the feature extraction is performed on the value of one of every two inputs. It can be seen that the dilation rate in TCN networks grows exponentially along with the increase in the number of layers, which makes the TCN network have a longer sensory field with only a few layers, thus outputting richer information features.

Compared with traditional CNN, TCN incorporates a residual linking module in model training, which aims to solve the problem of missing information that occurs when the number of TCN layers is high. When a convolutional operation is performed, this module works by superimposing the inputs and outputs to ensure that information is not lost. This module consists of two causally expanded convolutional layers and other related modules combined to form a residual module as the basic unit of the TCN network, the structure of which is shown in Fig 2.

In practical power load forecasting, the factors affecting power load include multiple variables such as ambient temperature, humidity, and electricity price. Therefore, this paper introduces the attention mechanism on the basis of TCN network to solve the complex dynamic dependency relationship between multiple variables. Its model structure is shown in Fig 3.

The first layer is the TCN layer, which captures the temporal information of the input multivariate time series and outputs the information representation ft at time $t$. The second layer is a CNN to perform feature extraction on the input temporal representation $f_t$. The second layer performs feature extraction on the input information representation $f_t$ at time $t$ through CNN. The input $f_t$ is reconstructed as a one-dimensional matrix $F = \{f_1, f_2, \ldots, f_{t-1}\}$, which is added to the CNN convolutional kernel for feature extraction, and the output is

$$F_{i,j}^C = C_{j,l} \sum_{l=1}^{w} F_{i,l-1}^C \qquad (13)$$

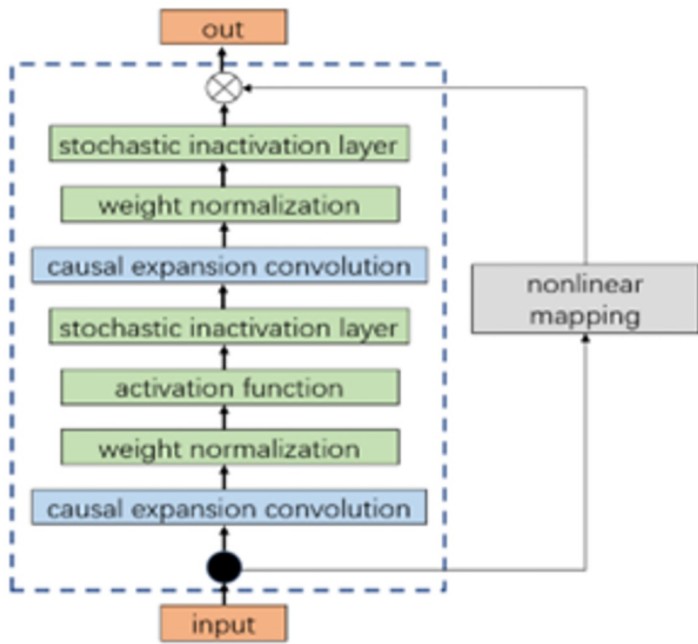

**Fig 2. Residual module unit structure.**

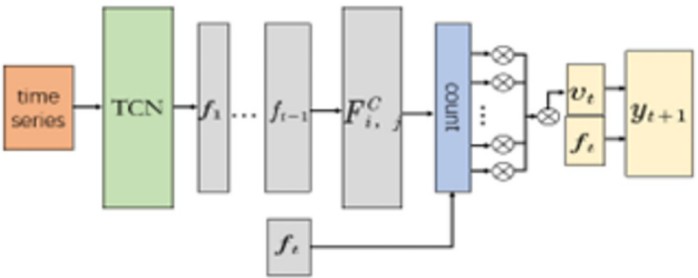

**Fig 3. TCN modeling with the introduction of an attention mechanism.**

Formula: $F_{i,j}^C$ is the convolution value of the $i$th row of vectors and the $j$th convolution kernel; $C$ is the convolution kernel; $l$ is the length of the time series.

The sigmoid function is chosen as the activation function in the last layer, and the output $F_{i,j}^C$ is weighted and the weight $\alpha_i$ is calculated. The obtained $\alpha_i$ is weighted to the $F_{i,j}^C$ row vector to obtain the new timing information $v_t$.

$$v_t = \sum_{i=1}^{t} F_i^C \alpha_i \tag{14}$$

Finally, $f_t$ is fused with $v_t$ to obtain the output value $y_{t+1}$ at the final $t+1$ time.

$$y_{t+1} = \varepsilon f_t v_t \tag{15}$$

Formula: $\varepsilon$ is the output coefficient.

## 2.4 Long- and short-term memory networks

Long Short-Term Memory Network (LSTM) is an improved form of Recurrent Neural Network (RNN), which effectively solves the problems of gradient explosion as well as gradient vanishing that often occur in RNNs [26–28]. Compared with the traditional RNN, LSTM has the ability to better capture and memorize long-term dependencies by introducing input gates, genetic gates, and output gates to control the transfer of information between them, and it can adaptively learn and capture the dependency information in sequence data. Its model structure is shown in Fig 4 below.

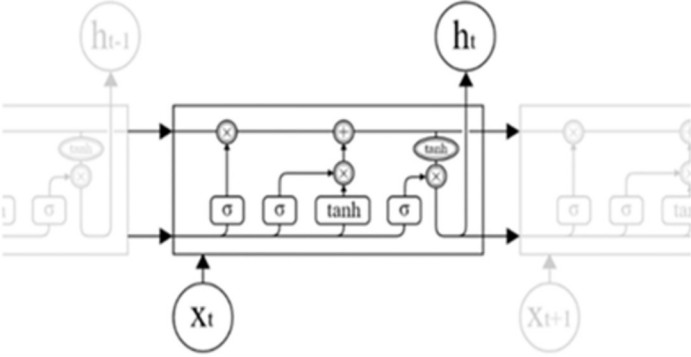

**Fig 4. LSTM network model diagram.**

The formulae for the variables in the LSTM network are shown below.

$$h_t = \tanh(c_t) * o_t \tag{16}$$

$$\sigma(x) = \frac{1}{1 + e^{-x}} \tag{17}$$

$$\tanh(x) = \frac{e^x - e^{-x}}{e^x + e^{-x}} \tag{18}$$

Formula: $h_t$ is the output of the current moment; $\sigma$ and tanh are the activation functions.

## 3 Electricity load forecasting model based on CEEMDAN and TCN-LSTM

Due to the influence of temperature, humidity, and electricity price and other factors, the power load data has a certain nonlinearity and non-stationarity. In order to reduce the influence of raw load data on future prediction, a combination of signal decomposition and deep learning neural network modeling is used to forecast electricity load. The model proposed in this paper introduces the concept of sample entropy on the basis of existing prediction models based on modal decomposition. By reorganizing the original signal sequences and reconstructing them into new sequences with significant differences in complexity, the error of multiple predictions is reduced, and the prediction efficiency is also improved. Then the reorganized data sequences are learned and predicted by combining TCN network and LSTM network, which fully captures the effective features and dependencies between the data and makes the prediction results more reliable.

The TCN-LSTM model in this paper gives full play to the features of TCN network and LSTM network by inputting the raw data into TCN network for feature information extraction, and inputting the extracted time-series features into LSTM network for mining and learning the dependency relationship between the data in order to achieve the best prediction effect. The TCN-LSTM model of this paper is shown in Fig 5 below.

The load forecasting model based on CEEMDAN and TCN-LSTM proposed in this paper is shown in Fig 6 below.

The model prediction steps can be roughly divided into four steps:

1. Data preprocessing of the load data by means of the CEEMDAN decomposition algorithm, which decomposes the input data into a number of linearly smooth eigenmode functions IMFs and a residual component Res.

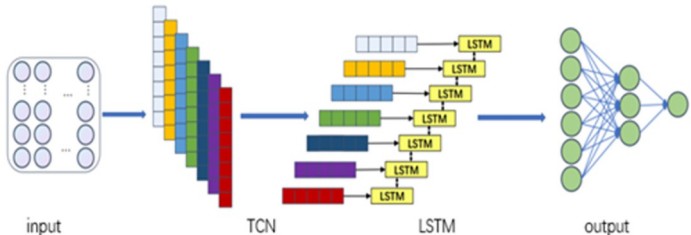

**Fig 5. TCN-LSTM model.**

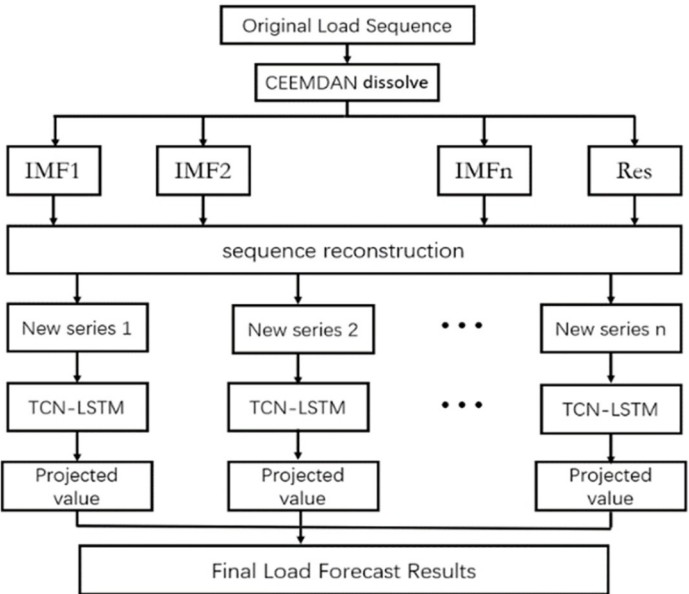

**Fig 6. CEEMDAN and TCN-LSTM based model structure.**

2. Calculate the value of each sequence after CEEMDAN decomposition by the sample entropy algorithm, and reorganize them into a new sequence combination based on their sample entropy value and the relationship between the sequences.

3. The combination of reorganized sequences is transmitted as input to the TCN-LSTM network model for learning and training, and the initial prediction values of each sequence are obtained.

4. Integrate the preliminary forecasts of the obtained individual sequences by linearly superimposing them to obtain the final load forecast.

## 4 Case study

In order to validate the effectiveness of the CEEMDAN and TCN-LSTM methods proposed in this paper, taking into account the period of the power load data and the influencing factors, this paper selects the comprehensive power load data of New South Wales, Australia, from January to December 2010, with a sampling interval of 1h, and collects the power load, dry bulb temperature, wet bulb temperature, dew-point temperature, humidity, and tariff data for 24h a day, and divides the training set, the validation set, and the test set according to the ratio of 8:1:1.

For the collected power load, dry bulb temperature, wet bulb temperature, dew point temperature and tariff data, which are of different orders of magnitude and units, in order to avoid oversaturation of neurons and thus affecting the feature analysis, the read data set is first normalized to map the data into the range [0,1], which is calculated as follows:

$$x_p = \frac{x - x_{min}}{x_{max} - x_{min}} \tag{19}$$

Formula: $x$ is the input data; $x_{\min}$ and $x_{\max}$ are the minimum and maximum values of the data; and $x_p$ is the normalized data.

Meanwhile, in order to evaluate the prediction performance of the model in this paper more intuitively and accurately, this paper selects the mean absolute error (MAE), the root mean square error (RMSE) and the mean absolute percentage error (MAPE) as the error evaluation indexes, which are calculated by the following formulas:

$$MAE = \sum_{i=1}^{n} |y_i - x_i| \tag{20}$$

$$RMSE = \sqrt{\frac{1}{n} \sum_{i=1}^{n} (y_i - x_i)^2} \tag{21}$$

$$MAPE = \frac{1}{n} \sum_{i=1}^{n} |\frac{y_i - x_i}{x_i}| \times 100\% \tag{22}$$

Formula: $n$ is the total sample size of electric load; $y_i$ is the predicted value of electric load; $x_i$ is the real value of electric load.

## 4.1 CEEMDAN sequence decomposition

In order to improve the prediction accuracy, the CEEMDAN algorithm was used to process the normalized data. The CEEMDAN parameters were set: the standard deviation of the added Gaussian white noise was 0.2 and the number of added noise was 300.

As shown in Fig 7 below, after CEEMDAN decomposition, the original time series is decomposed into 10 IMF components and one residual component Res. The IMF components are arranged in order of frequency from high to low, and each component is relatively stable without modal aliasing.

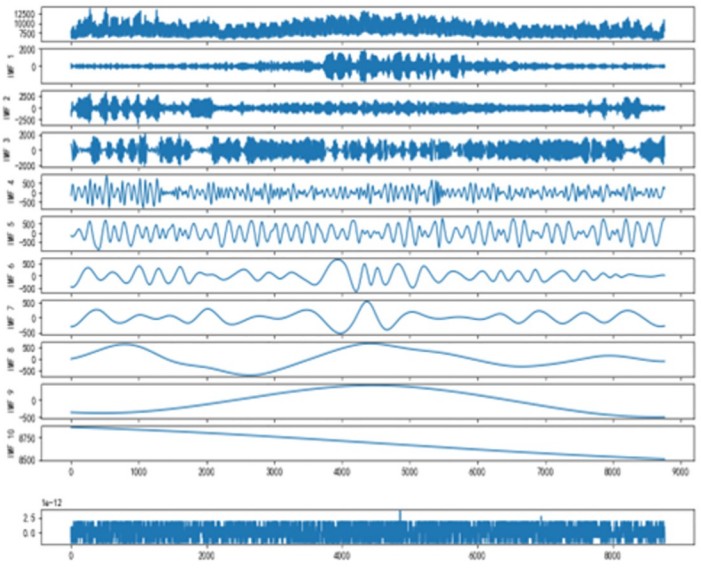

**Fig 7. IMF components after CEEMDAN decomposition.**

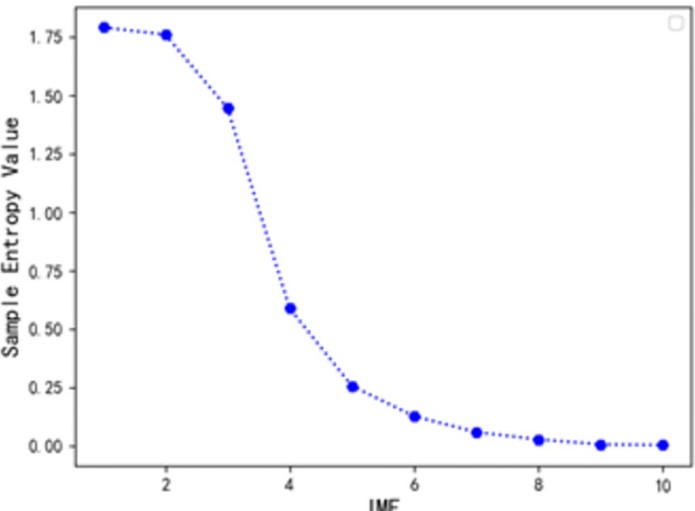

**Fig 8. Sample entropy values for each IMF component.**

After CEEMDAN decomposition, the frequency characteristics of different IMF components are different. In order to better feature extraction of modal components, this paper adopts the sample entropy theory to analyze the complexity of each IMF component, and the sample entropy results obtained for each IMF component are shown in Fig 8 and Table 1.

From Fig 8 and Table 1, it can be seen that the sample entropy values of IMF1 and IMF2 are both greater than 1.75 and the sample entropy values of these two subsequences are very close to each other, indicating that the complexity of these two sequences are relatively high and the probability of generating a new pattern is basically the same, so IMF1 and IMF2 are superimposed as a new subsequence to be trained and predicted, and call the new subsequence NIMF1; the sample entropy value of IMF3 is different from that of other The sample entropy value of IMF3 is closer to the sample entropy value of the other IMF components and its complexity is higher, so IMF3 is used as a separate subsequence for training and prediction, and the new subsequence is called NIMF2; the sample entropy value of IMF4 is closer to the sample entropy value of the residual component Res, and the difference is only 0.017, so IMF4 is superimposed on Res as a new subsequence for training and prediction and the new subsequence is called NIMF3. The sample entropy values of IMF5 and IMF6 are very close to each other, which means that the complexity of these two sequences is relatively high and the probability of generating a new pattern is basically the same, so IMF5 and IMF6 are superimposed as a new subsequence to be trained and predicted and called NIMF4; the sample entropy values of these four IMF components IMF7~IMF10 are relatively close to each other, so the four sequences are superimposed to get the new NIMF4; the sample entropy values of these four IMF components are relatively close to each other. These 4 sequences are superimposed to get

**Table 1. Sample entropy values for each component.**

| IMF | IMF1 | IMF2 | IMF3 | IMF4 | IMF5 | IMF6 |
|---|---|---|---|---|---|---|
| SE | 1.795 | 1.762 | 1.459 | 0.580 | 0.253 | 0.096 |
| IMF | IMF7 | IMF8 | IMF9 | IMF10 | Res | |
| SE | 0.042 | 0.026 | 0.024 | 0.015 | 0.563 | |

**Table 2. Reconstructed subsequences.**

| new | NIMF1 | NIMF2 | NIMF3 | NIMF4 | NIMF5 |
|---|---|---|---|---|---|
| previous | 1,2 | 3 | 4,Res | 5, 6 | 7~10 |

the new subsequence NIMF5 for training and prediction. The reconstructed sequence is shown in Table 2 below.

## 4.2 Analysis of projected results

The reorganized sequences NIMF1~NIMF5 are constructed according to the sliding window iterative prediction, and its sliding window size is set to 24, i.e., the 24 data of the previous 24h are taken as the features, while the power load data of the next hour are taken as the labels. By feeding the training set into the TCN-LSTM network model of this paper for training, while the test set is used for the prediction of the next hour and the validation set is used to verify the accuracy of the prediction.

The learning rate of the TCN-LSTM network is set to 0.01, the regularization dropout parameter is 0.1, the convolution kernel size is 3, the activation function is a sigmoid function, and the number of training rounds is 50. Fig 9 shows the loss value plot of the reconstructed sequence after training by the TCN-LSTM network model.

As can be seen from Fig 9, this TCN-LSTM model converges faster at the same time the model does not have overfitting phenomenon, which indicates that the accuracy and generalization ability of the model is better.

To further validate the performance of CEEMDAN-TCN-LSTM network and its accuracy in power load forecasting, based on the same training set, test set and validation set, respectively, SVR algorithm, TCN network model, LSTM network model, TCN-LSTM network model, and EMMD-TCN-LSTM (hereafter E-T-L) hybrid model are conducted with comparative experiments were conducted. In the experiment, 300 sampling points are selected to

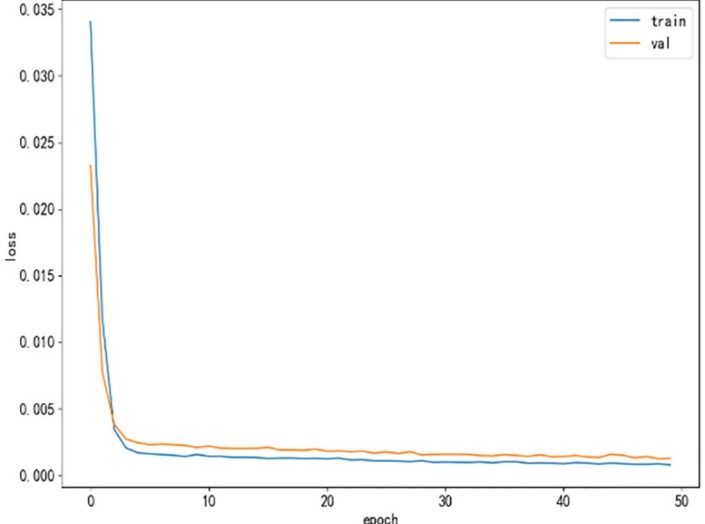

**Fig 9. TCN-LSTM network loss value.**

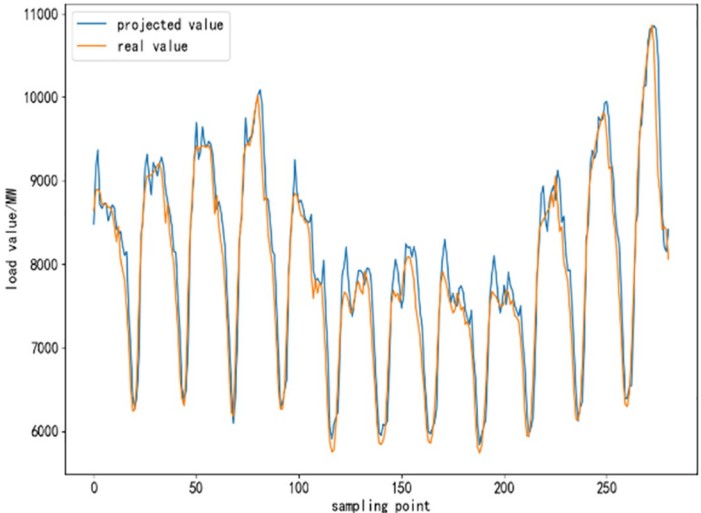

**Fig 10. SVR prediction results.**

compare the differences and similarities between the predicted values and the real values after the prediction of each model, and the results of the comparison experiment are shown as follows.

From Fig 10 to Fig 15, it can be seen that the CEEMDAN-LSTM-CNN model better grasps the law of the load change ground, and the prediction curves fit the actual load curves better compared to SVR, LSTM, TCN, TCN-LSTM, and EMMD-TCN-LSTM, and the fitting around the extreme points of the sequence is better.

The MAE, RMSE, and MAPE values as well as the prediction accuracies of each model prediction are given in Table 3. As can be seen from Table 3, compared with the single deep

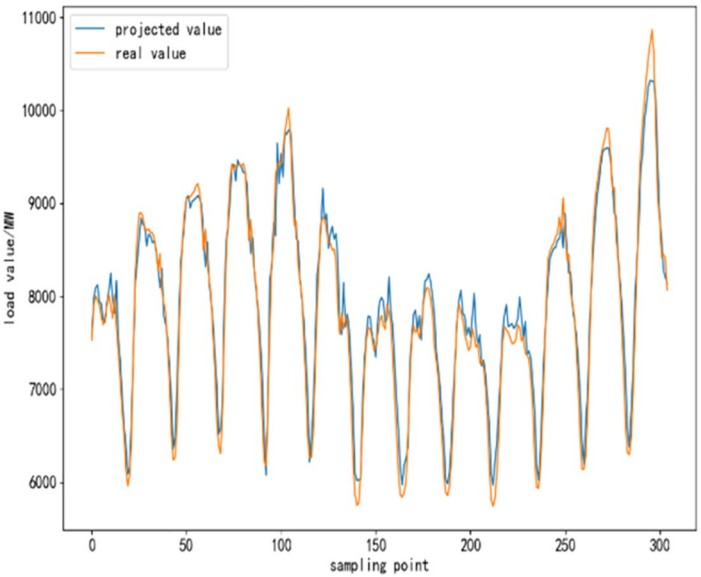

**Fig 11. LSTM prediction results.**

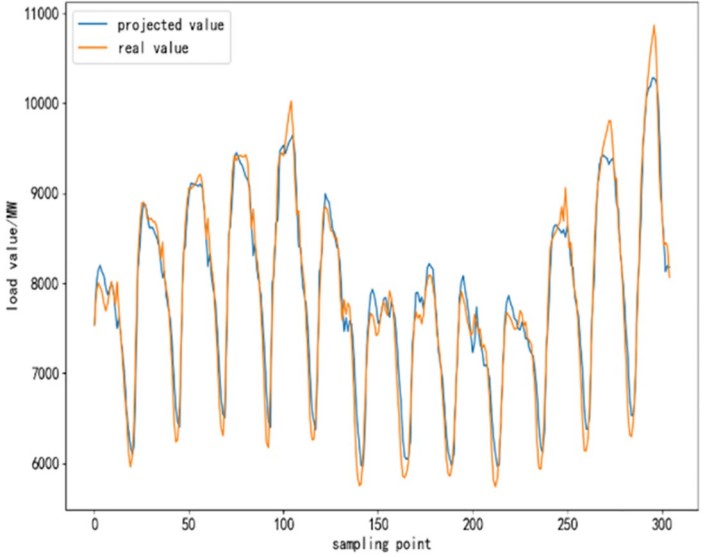

**Fig 12. TCN prediction results.**

learning prediction models SVR, TCN, and LSTM, the evaluation indexes of the models proposed in this paper are significantly lower, and the prediction accuracies are also improved. Secondly, on the current load data, the TCN-LSTM model prediction effect is also better than the single TCN model and LSTM model, which indicates that the TCN model can more fully exploit the time-order characteristics of the time-load sequence ground. By comparing the TCN-LSTM model with the TCN-LSTM model after CEEMDAN decomposition in this paper, it can be seen that its MAE, RMSE and MAPE values are reduced by 63.286, 80.34 and

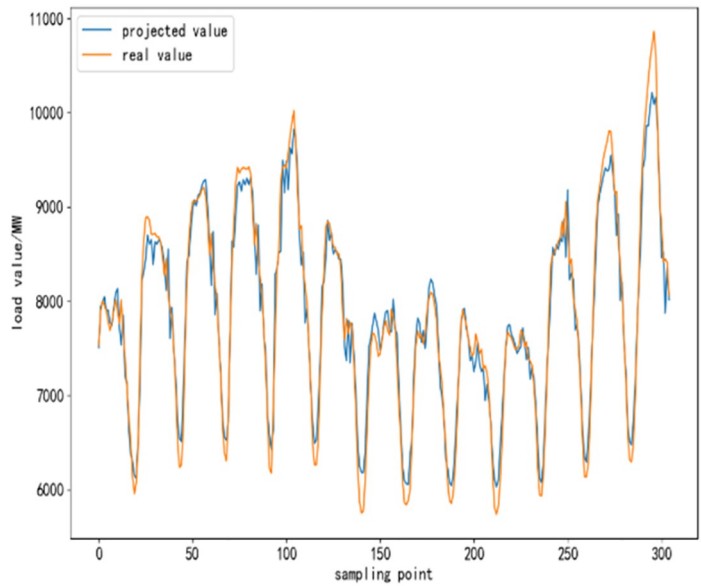

**Fig 13. TCN-LSTM prediction results.**

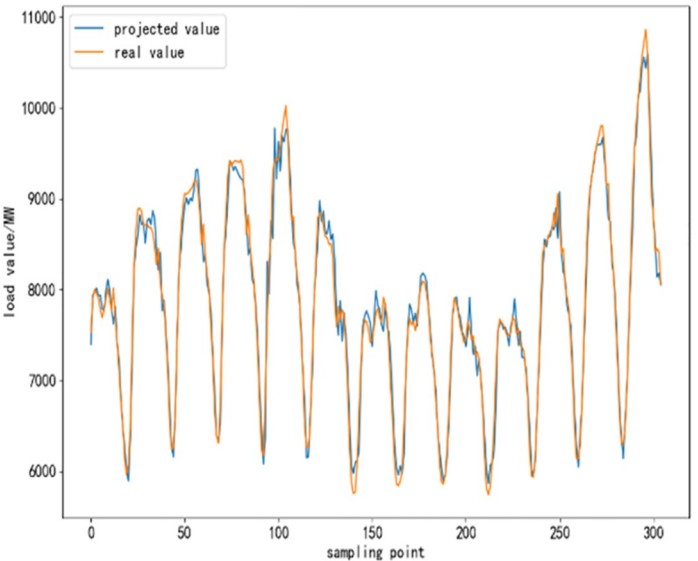

**Fig 14. EMMD-TCN-LSTM prediction results.**

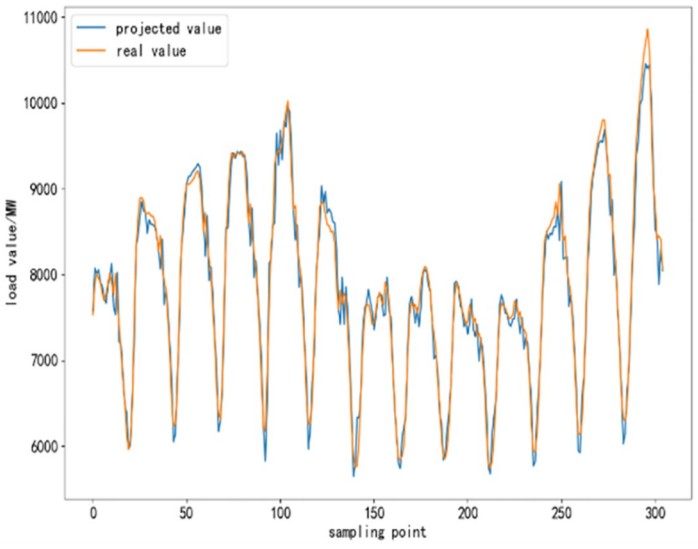

**Fig 15. CEMMD-TCN-LSTM prediction results.**

**Table 3. Evaluation indexes and accuracy of each model prediction.**

| Mould | MAE | RMSE | MAPE | Accuracy |
|---|---|---|---|---|
| SVR | 268.930 | 337.446 | 8.74% | 91.26% |
| LSTM | 190.105 | 255.461 | 6.33% | 93.67% |
| TCN | 193.336 | 237.626 | 4.76% | 95.24% |
| TCN-LSTM | 174.226 | 226.054 | 4.17% | 95.83% |
| E-T-L | 151.338 | 189.571 | 3.26% | 96.74% |
| essay | 110.94 | 145.714 | 1.86% | 98.14% |

2.31% respectively, which indicates that the prediction performance of the model is further improved after linear smoothing of the time-load sequence by CEEMDAN. Improvement. Meanwhile, the algorithm in this paper reduces the error accumulation caused by multiple predictions after introducing sample entropy for sequence complexity analysis and time series reconstruction, which also effectively improves the prediction accuracy of the CEEMDAN-TCN-LSTM model in this paper.

## 5 Concluding remarks

In this paper, a short-term power load forecasting method based on CEEMDAN decomposition and TCN-LSTM network combination model is proposed for power load forecasting. The method firstly decomposes the power load data into sequences by CEEMDANJIAN, then reorganizes each sequence after decomposition into a new linear smooth sequence according to the sample entropy value, and then the reorganized sequences are de-trained with TCN-LSTM network that introduces the attention mechanism and forecasts the future power load. The algorithm in this paper is compared with existing signal decomposition and deep learning algorithms and the following conclusions are obtained:

1. Compared with the traditional signal decomposition method, the CEEMDAN decomposition method proposed in this paper decomposes the time series data more completely and better reduces the error of sequence reconstruction, and improves the prediction accuracy to some extent.

2. In this paper, sample entropy is introduced to analyze the complexity of each subsequence after CEEMDAN decomposition, and reconstructed into a new linear smooth sequence according to the sample entropy value of the individual sequences, which further improves the accuracy of prediction.

3. In this paper, the attention mechanism is introduced into TCN spatio-temporal convolutional network, so that TCN can better solve the interdependence between complex variables, and enhance the feature mining as well as learning ability of TCN network.

4. In this paper, a combined network model of TCN and LSTM is constructed, which is seen over the TCN network and LSTM network again, avoiding the inadequate extraction of potential features of the time series data by a single neural network, and thus improving the overall load prediction accuracy.

5. Compared with other common forecasting algorithms, this paper's algorithm can more accurately predict the changes in actual load data and perform better in the evaluation indexes, which can provide a certain reference for the power sector's electric power forecasting as well as electric energy scheduling.

## Supporting information

**S1 Data.**
(CSV)

## Author Contributions

**Conceptualization:** Liu Chen Nan.

**Data curation:** Cheng Hao.

**Formal analysis:** Luo Heng, Cheng Hao.

**Investigation:** Luo Heng.

**Methodology:** Cheng Hao.

**Project administration:** Liu Chen Nan.

**Software:** Luo Heng.

**Supervision:** Luo Heng.

**Writing – original draft:** Cheng Hao.

**Writing – review & editing:** Cheng Hao.

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
