## [Decision Letter · Decision Letter 0]

20 Mar 2024

PONE-D-24-07774Load Forecasting Method Based on CEEMDAN and TCN-LSTMPLOS ONE

Dear Dr. 成,

Thank you for submitting your manuscript to PLOS ONE. After careful consideration, we feel that it has merit but does not fully meet PLOS ONE’s publication criteria as it currently stands. Therefore, we invite you to submit a revised version of the manuscript that addresses the points raised during the review process. Please check the comments from the reviewers, especially the ones related to the dataset and its availability, as well as to the LTSM diagram that needs to be corrected.

We look forward to receiving your revised manuscript.

Kind regards,

Tomo Popovic, Ph.D.

Academic Editor

PLOS ONE

Journal Requirements:

4. Please amend the manuscript submission data (via Edit Submission) to include authors Cheng, Liu Chen Nan.

5. Please upload a copy of Figure 15, to which you refer in your text on page 18. If the figure is no longer to be included as part of the submission please remove all reference to it within the text

Additional Editor Comments (if provided):

Please check the comments from the authors, especially related to the dataset and LTSM diagram that needs to be corrected.

Reviewers' comments:

Reviewer's Responses to Questions

**Comments to the Author**

1. Is the manuscript technically sound, and do the data support the conclusions?

Reviewer #1: Yes

Reviewer #2: Yes

2. Has the statistical analysis been performed appropriately and rigorously? 

Reviewer #1: Yes

Reviewer #2: Yes

3. Have the authors made all data underlying the findings in their manuscript fully available?

Reviewer #1: No

Reviewer #2: No

4. Is the manuscript presented in an intelligible fashion and written in standard English?

Reviewer #1: Yes

Reviewer #2: Yes

5. Review Comments to the Author

Reviewer #1: The data supports the conclusions, and experiments, including statistical analysis, are conducted rigorously, with appropriate controls, replication, and sample sizes. Based on the data presented, the conclusions are drawn appropriately.

The manuscript presents a novel approach to load forecasting, combining advanced deep learning methods (Temporal Convolutional Network and Long-Short-Term Memory) with an innovative method for decomposing raw data (Completely Integrated Empirical Modal Decomposition). This unique combination forms the basis of technically sound scientific research.

One limitation of the manuscript is that the authors should have provided all the data underlying the findings. This data was not fully available as part of the manuscript or its supporting information or deposited in a public repository. This lack of data availability could limit the reproducibility of the research.

The manuscript is clear, correct, unambiguous, presented in an intelligible fashion, and written in standard English. However, it is not formatted correctly, as line spacings in regular text are not equal everywhere. Some figures are not correctly placed and overlap with their description, or their descriptions overlap with regular text. Also, it remains unclear whether figures 1-6 and equations are used from the literature.

Reviewer #2: Scientifically, the paper is very sound and explores the most modern approaches to the problem at hand. Results are very promising and it would be very interesting to see these methods applied to electricity price forecasting.

In the data availability statement, it is said that all the data is available without restriction, but there is no information on where the data can be found, neither in the statement nor in the text.

The citation style is inconsistent. Sometimes it is in superscript, sometimes it is broken into 2 lines. [20] is quoted as "literature" instead of author names.

Moreover, the LSTM diagram is wrong. Rounded corners imply pointwise operations, and rectangles imply neural (trainable) parts of the network. There are two tanh elements in the diagram and they are drawn as if they were the same - but they are not. One of them is pointwise tanh, and other one is neural, and should be drawn as such (see attached image).

6. PLOS authors have the option to publish the peer review history of their article (what does this mean?). If published, this will include your full peer review and any attached files.

Reviewer #1: No

Reviewer #2: **Yes: **Milutin Pavićević

---

## [Author Response · Author response to Decision Letter 0]

24 Mar 2024

Dear reviewers，

Thank you for reviewing my submitted thesis. After reading the review comments that you have made，I have revised my thesis in response to the comments you have made. The specific modifications are listed below:

First，I have uploaded the datasets used in this experiment and thesis for reference.

Secondly，I've made changes to the line spacing in the body text and the placement of the charts.

Thirdly, the citation for reference 20 in the text has been changed to the author's name.

Fourthly, the graphs of the LSTM in the text have been modified to those provided in the Appendix.

I have made changes in response to the review comments and I will resubmit the revised paper. Thank you again for your contribution to the review of my thesis.

Your sincerely,

Luo Heng

Cheng Hao

Liu Chen Nan

Suzhou , China

---

## [Decision Letter · Decision Letter 1]

16 Apr 2024

PONE-D-24-07774R1Load Forecasting Method Based on CEEMDAN and TCN-LSTMPLOS ONE

Dear Dr. 洛,

Thank you for submitting your manuscript to PLOS ONE. After careful consideration, we feel that it has merit but does not fully meet PLOS ONE’s publication criteria as it currently stands. Therefore, we invite you to submit a revised version of the manuscript that addresses the points raised during the review process.

We look forward to receiving your revised manuscript.

Kind regards,

Tomo Popovic, Ph.D.

Academic Editor

PLOS ONE

Journal Requirements:

**Additional Editor Comments:**

Plese check the comments and feedback from the reviewers.

Reviewers' comments:

Reviewer's Responses to Questions

**Comments to the Author**

1. If the authors have adequately addressed your comments raised in a previous round of review and you feel that this manuscript is now acceptable for publication, you may indicate that here to bypass the “Comments to the Author” section, enter your conflict of interest statement in the “Confidential to Editor” section, and submit your "Accept" recommendation.

Reviewer #1: (No Response)

Reviewer #2: All comments have been addressed

2. Is the manuscript technically sound, and do the data support the conclusions?

Reviewer #1: Yes

Reviewer #2: Yes

3. Has the statistical analysis been performed appropriately and rigorously? 

Reviewer #1: Yes

Reviewer #2: Yes

4. Have the authors made all data underlying the findings in their manuscript fully available?

Reviewer #1: Yes

Reviewer #2: Yes

5. Is the manuscript presented in an intelligible fashion and written in standard English?

Reviewer #1: Yes

Reviewer #2: Yes

6. Review Comments to the Author

Reviewer #1: Formatting issues from the previous review have not been addressed properly, so I suggest minor revision again. However, underlying data is fully available in the revised version.

Reviewer #2: All the concerns have been addressed. Please correct the format of citations. There are still some citations ileft n superscript.

7. PLOS authors have the option to publish the peer review history of their article (what does this mean?). If published, this will include your full peer review and any attached files.

Reviewer #1: No

Reviewer #2: No

---

## [Author Response · Author response to Decision Letter 1]

18 Apr 2024

Dear reviewers，

Thank you for reviewing my submitted thesis. After reading the review comments that you have made，I have revised my thesis in response to the comments you have made. The specific modifications are listed below:

First,I have checked the journal's template for papers and changed all the first-level headings in the paper to bold, 18-point font, and the second-level headings in the paper to bold, 16-point font.

Secondly, The font size of the text has been changed to a uniform 12-point font size.

Thirdly ,I've changed the spacing of the text and the placement of the charts again.

Finally, All references cited in the main text have been changed to superscript citations.

I have made changes in response to the review comments and I will resubmit the revised paper. Thank you again for your contribution to the review of my thesis.

Your sincerely,

Luo Heng

Cheng Hao

Liu Chen Nan

Suzhou , China

---

## [Editor Report · Decision Letter 2]

1 May 2024

Load Forecasting Method Based on CEEMDAN and TCN-LSTM

PONE-D-24-07774R2

Dear Dr. 洛,

We’re pleased to inform you that your manuscript has been judged scientifically suitable for publication and will be formally accepted for publication once it meets all outstanding technical requirements.

Kind regards,

Tomo Popovic, Ph.D.

Academic Editor

PLOS ONE